# Transoral Robotic Surgery for Oropharyngeal Squamous Cell Carcinoma of the Tonsil versus Base of Tongue: A Systematic Review and Meta-Analysis

**DOI:** 10.3390/cancers14153837

**Published:** 2022-08-08

**Authors:** Nicolas S. Poupore, Tiffany Chen, Shaun A. Nguyen, Cherie-Ann O. Nathan, Jason G. Newman

**Affiliations:** 1Department of Otolaryngology—Head and Neck Surgery, Medical University of South Carolina, Charleston, SC 29425, USA; 2Department of Otolaryngology—Head and Neck Surgery, Louisiana State University Health Sciences Center, Shreveport, LA 71103, USA

**Keywords:** oropharyngeal squamous cell carcinoma, transoral robotic surgery, tonsil, base of tongue, positive margin, recurrence, postoperative hemorrhage

## Abstract

**Simple Summary:**

Transoral Robotic Surgery (TORS) has become widely adopted for the surgical removal of oropharyngeal squamous cell carcinoma (OPSCC). However, it is currently unknown if TORS has equal efficacy and outcomes in patients with tonsillar or base of tongue (BOT) OPSCC. Therefore, we performed a systematic review, including articles describing the surgical management of OPSCC with TORS that compared margin status, complications, and recurrence between tonsil and BOT. BOT OPSCC had a higher rate of positive margins compared to tonsillar OPSCC. However, no differences were seen in the recurrence or postoperative hemorrhage rates of BOT and tonsillar OPSCC. While a higher rate of positive margins was seen in BOT OPSCC when compared to tonsillar OPSCC, this did not translate to a higher recurrence rate in the BOT group. Future research on which subset of patients with BOT is more likely to have positive margins is warranted to improve the utility of TORS further.

**Abstract:**

Transoral Robotic Surgery (TORS) has become widely adopted for the surgical removal of oropharyngeal squamous cell carcinoma (OPSCC), with the most common locations being in the tonsil and base of tongue (BOT). However, it is currently unknown if TORS has equal efficacy and outcomes in patients with tonsillar or BOT OPSCC. Therefore, the aim of this study was to compare the margin status and recurrence rates of tonsillar and BOT OPSCC after TORS. Per PRISMA guidelines, PubMed, Scopus, and CINAHL were systematically searched from inception to 2/28/2022. Articles including the surgical management of OPSCC with TORS that compared margin status, complications, and recurrence between tonsil and BOT were included. Meta-analyses of proportions and odds ratios were performed. A total of 28 studies were included, comprising 1769 patients with tonsillar OPSCC and 1139 patients with BOT OPSCC. HPV positivity was seen in 92.3% of tumors. BOT OPSCC had a higher rate of positive margins compared to tonsillar OPSCC (28.1% [95%CI 15.1–43.3] vs. 7.5% [95%CI 3.3–13.3]). No differences were seen in recurrence between BOT and tonsillar OPSCC (OR 1.1 [95%CI 0.8–1.5], *p* = 0.480). In addition, no differences in postoperative hemorrhage were seen between tonsillar and BOT OPSCC (10.7% [95%CI 6.1–16.5] vs. 8.8% [95% CI 1.5–21.3]). While a higher rate of positive margins was seen in BOT OPSCC when compared to tonsil OPSCC, this did not translate to a higher recurrence rate in the BOT group. Future research on which subset of patients with BOT is more likely to have positive margins is warranted to improve the utility of TORS further.

## 1. Introduction

The incidence of oropharyngeal squamous cell carcinoma (OPSCC) has consistently risen over the past 40 years, secondary to an increase in human papillomavirus (HPV)-positive tumors [1,2,3]. OPSCC used to primarily be associated with tobacco and alcohol use, which has been shown to lead to a more aggressive disease pathology compared to HPV-associated OPSCC [4,5,6]. Secondary to the changing etiologies and pathogeneses of OPSCC, different therapeutic approaches and techniques, such as de-escalation strategies, are currently being explored to improve survival and long-term outcomes in patients with OPSCC [4,7]. Evidence shows that upfront transoral robotic surgery (TORS) is as effective as chemoradiotherapy, and it has been widely adopted for the treatment of T1 and T2 primary OPSCC [8]. While TORS was initially used in the surgical resection of tonsillar OPSCC, it has now advanced to also removing base of tongue (BOT) tumors [9,10,11].

However, there is currently limited and conflicting evidence on if there are varying surgical outcomes between the two most common sites of OPSCC: the tonsil and BOT [10]. One retrospective study noticed difficulty gaining proper exposure in BOT OPSCC compared to tonsillar OPSCC [12]. The authors also felt that the cryptic lingual tonsillar mucosa compared to the smooth mucosa of the tonsil made it more challenging to identify the extent of the tumor, particularly at the deep margin, which can be the hardest to clear [12,13]. While postoperative positive margins are crucial for the decision to perform adjuvant therapy, there is conflicting evidence on its utility when considering long-term survival [14,15]. One study found that patients requiring more intraoperative attempts to achieve a negative margin had a higher rate of death due to cancer [16], but two other studies did not find any changes in survival when comparing patients with positive margins to negative margins in HPV-positive OPSCC after TORS [17,18].

It is currently unknown if surgical outcomes differ between tonsillar and BOT OPSCC after TORS. Therefore, we performed a systematic review of the literature to analyze tumor margin status and recurrence rates after TORS. Secondarily, we also aimed to see if complication rates differed between the subsites. Our initial hypothesis was that BOT OPSCC would have increased rates of positive margins after TORS, which would translate to higher rates of recurrence when compared to patients with tonsillar OPSCC. We also hypothesized that BOT tumors would have more complications secondary to more technically challenging removals and attempts at negative margins.

## 2. Materials and Methods

### 2.1. Search Criteria

This systematic review was performed following the Preferred Reporting Items for Systematic Reviews and Meta-Analyses (PRISMA) guidelines [19]. Two authors (N.S.P. and T.C.) developed search strategies for the following databases: PubMed (National Library of Medicine, National Institutes of Health); Scopus (Elsevier); and CINAHL (EBSCO) to identify studies for inclusion. These strategies used a combination of subject headings (e.g., Medical Subject Headings [MeSH] in PubMed) and keywords for the concepts oropharyngeal squamous cell carcinoma and TORS. The PubMed search strategy was developed first and then used to design the search strategies for the other two databases; MeSH terms were replaced with appropriate subject headings and similar keywords when available. The search strategies are detailed in Appendix B. The databases were searched from inception to 28 February 2022. The reference lists of relevant articles and citing articles were manually searched to confirm the search strategy and identify additional articles. All the articles were uploaded to the review management software Covidence (Veritas health innovation Ltd., Melbourne, Australia) and screened for relevance.

### 2.2. Selection Criteria

Studies that described cases of OPSCC treated with TORS were considered for inclusion. Double- or single-blinded randomized controlled trials, double- or single-blinded randomized comparison trials, non-randomized controlled trials, and prospective or retrospective observational studies were considered for inclusion. Specifically, cases must have described outcome data pertaining to the TORS surgery to be considered for inclusion. For this review, the outcomes of postoperative margin status, hemorrhage, and locoregional or metastatic recurrence were the primary outcomes of interest. Studies were excluded if they did not detail these three outcome variables based on either the tonsillar or BOT subsite. Furthermore, studies were excluded if outcomes were not specifically after TORS, such as with transoral or laser surgery. Studies that combined their outcome data from OPSCC with other malignancies were excluded. The remaining exclusion criteria included cancer database studies, case reports with less than four tonsillar or BOT OPSCC patients, non-English studies, and non-human studies.

Titles and abstracts were first independently screened by two reviewers (N.S.P. and T.C.) to identify the records that met the inclusion criteria. Disagreements were resolved in discussion with a third reviewer (S.A.N.). Next, the full texts of the selected records were independently assessed by both N.S.P. and T.C. to find which articles met all the inclusion and exclusion criteria necessary to be included in the final analysis. Any conflicts were resolved by S.A.N. The level of evidence for each selected article was evaluated with the Oxford Center for Evidence-Based Medicine [20]. The risk of bias for each selected report was assessed according to the Cochrane Handbook for Systematic Reviews of Interventions Version 6.0 [21]. The Risk of Bias in Non-Randomized Studies – of Interventions (ROBINS-I) tool was utilized because all the final selected studies were non-randomized studies [22]. The consistency of the risk of bias assessment was checked by the two authors (N.S.P. and T.C.) performing a pilot assessment on three studies and comparing for accuracy. Both then performed an independent risk assessment on the remaining studies. All disagreements were resolved by discussion with a third reviewer (S.A.N.). The risk of bias items included the following: bias due to confounding, bias in the selection of participants for the study, bias in the classification of interventions, bias due to deviations from the intended interventions, bias due to missing data, bias in the measurement of outcomes, and bias in the selection of the reported results. The risk of bias for each aspect was graded as low, unclear, or high.

### 2.3. Data Extraction

Two reviewers (N.S.P. and T.C.) independently extracted the data and compared for accuracy. The author; year of publication; country of study; demographics, including age and gender; and smoking status were recorded. In addition, disease characteristics for the entire cohort of patients were collected, including p16/HPV status, T-stage, N-stage, perineural invasion, extranodal extension, lymphovascular invasion, and extrascapular spread. The presence of neck dissection and adjuvant therapy (radiotherapy or chemoradiotherapy) was also recorded. The following outcomes specific to tonsillar or BOT subsite were extracted: the number of positive margins, locoregional and metastatic recurrence, and postoperative hemorrhage. Locoregional and metastatic recurrence was combined into the variable total recurrence. If a study did not specify whether the recurrence was locoregional or metastatic, those cases were counted only under total recurrence.

### 2.4. Statistical Analysis

Meta-analyses of odds ratios (comparison of locoregional recurrence, metastatic recurrence, and total recurrence) between BOT and tonsil were performed with Cochrane Review Manager (RevMan) version 5.4 (The Cochrane Collaboration 2020). A meta-analysis of proportions was performed using MedCalc 19.6 (MedCalc Software Ltd., Ostend, Belgium; https://www.medcalc.org; accessed on 4 May 2020). The pooled prevalence rate of positive margin and hemorrhage for BOT and tonsil were expressed as a percentage with their 95% confidence intervals (CI). Each measure was weighted according to the number of patients affected. The weighted-summary proportion was calculated by the Freeman–Tukey transformation [23]. Heterogeneity among the studies was assessed using χ^2^ and I^2^ statistics. I^2^ < 50% indicated acceptable heterogeneity, and, therefore, the fixed-effects model was used; otherwise, the random-effects model was performed. In addition, a comparison of weighted proportions was performed to compare the prevalence rates of positive margin and hemorrhage between BOT and tonsil. Finally, Egger’s tests with funnel plots were performed to further assess the risk of publication bias [24,25]. In a funnel plot, the treatment effect is plotted on the horizontal axis, and the standard error is plotted on the vertical axis. The vertical line represents the summary estimated and is derived using a fixed-effect meta-analysis. Two diagonal lines represent (pseudo) 95% confidence limits (effect ± 1.96 SE) around the summary effect for each standard error on the vertical axis. These show the expected distribution of studies in the absence of heterogeneity or selection bias. In the absence of heterogeneity, 95% of the studies should lie within the funnel defined by these diagonal lines. Potential publication bias was evaluated by a visual inspection of the funnel plot (as bias results in asymmetry of the funnel plot), and Egger’s test, which statistically examines this asymmetry. A *p*-value of <0.05 was considered to indicate a statistically significant difference for all the statistical tests.

## 3. Results

### 3.1. Search Results and Study Characteristics

The search strategies yielded 1386 unique articles, with title and abstract screening excluding 1036 articles. A full-text review of the remaining studies excluded 322 articles, leaving 28 remaining articles for inclusion in the final data extraction and analysis. Figure 1 shows the PRISMA flowchart, which details the entire search process.

A critical appraisal of the studies indicated an acceptably low risk of bias for the majority of those that were included (Figure 2).

Potential sources of bias were most pronounced in bias in the selection of participants and bias in the selection of the reported results. A funnel plot with Egger’s test suggested little publication bias, as all the studies were within the funnel with no asymmetry (0.4, 95% CI-1.0–1.8, *p* = 0.568) (Appendix A).

Articles selected for inclusion were level 4 studies based on the Oxford Level of Evidence and were published between 2009 and 2022 (Table 1).

### 3.2. Patient Characteristics

A total of 2908 patients were included from all 28 studies (Table 1). The tonsillar OPSCC group had 1769 patients, and the BOT OPSCC group had 1139 patients. All patients’ mean and median age ranged from 54.0 years to 63.7 years, with most patients being male (range 62.9% to 95.7%). Most included patients did not have advanced disease (T3 and T4 range 0.0% to 26.1%), but there was a wide range of adjuvant radiotherapy (0.0% to 100.0%) and chemoradiotherapy (0.0% to 100%). The mean and median follow-up ranged from 10.5 to 48.4 months for the included studies.

### 3.3. Margin Status

A total of 12 studies included extractable data on margin status after resection using TORS [12,26,27,28,29,30,31,32,33,34,35,36]. BOT OPSCC had a statistically significant higher rate of positive margins when compared to tonsillar OPSCC (28.1% [95% CI 15.1–43.3] vs. 7.5% [95% CI 3.3–13.3]). Both forest plots are shown in Figure 3A,B.

### 3.4. Recurrence

The recurrence of OPSCC after TORS was analyzed by subsite in 13 studies [37,38,39,40,41,42,43,44,45,46,47,48,49]. There was no statistically significant difference in the odds of total recurrence between tonsillar and BOT OPSCC (1.1 [95%CI 0.8–1.5], *p* = 0.480) (Figure 4).

Furthermore, no differences were seen in the odds of recurrence when separated by locoregional (1.2 [95% CI 0.8–1.8], *p* = 0.500) and metastatic recurrence (0.8 [95% CI 0.4–1.6], *p* = 0.600). These forest plots are shown in Figure 5 and Figure 6.

### 3.5. Hemorrhage

Postoperative hemorrhage was described by subsite in nine articles for this review [26,27,28,32,34,44,50,51,52]. There was no statistical difference between the hemorrhage rates in patients with tonsillar or BOT OPSCC treated with TORS. (10.7% [95% CI 6.1–16.50] vs. 8.8% [95% CI 1.5–21.3]) (Figure 7A,B).

## 4. Discussion

There is a paucity of literature comparing the postoperative outcomes after TORS by subsite, particularly tonsil and BOT. Therefore, we performed this systematic review and meta-analysis to compare the margin, recurrence, and postoperative hemorrhage rates between patients with tonsillar and BOT OPSCC. Notably, patients with BOT OPSCC had higher rates of positive margins after TORS, but contrary to our initial hypothesis, this did not translate to a higher locoregional or metastatic recurrence rate. Furthermore, postoperative hemorrhage rates were not noted to be different between these two subgroups of patients undergoing TORS.

In our included studies, patients with BOT OPSCC had a higher proportion of positive postoperative margins compared to patients with tonsillar OPSCC (28.1% vs. 7.5%). In OPSCC, the positive margin rate for all subsites after TORS has been cited as between 16.9% and 21.2% [53,54,55]. The utility of TORS regarding improvement in margin status is inconsistent. One systematic review that combined outcomes from TORS, transoral laser microsurgery (TLM), and transoral conventional surgery found a lower positive margin rate of 7.8%, with the tonsil and BOT subsites not being factors in margin status [56]. However, another retrospective study of the National Cancer Database (NCDB) found that TORS was associated with a lower likelihood of positive margins for all sites compared to non-robotic procedures, but not TLM [57]. When considering outcomes after TORS, another NCDB study by Hanna et al. found the rate of BOT OPSCC positive margins was higher than tonsillar OPSCC, but this was nonsignificant [55]. Interestingly, Hanna et al. and another more recent NCDB review from Oliver et al. found that high volume cancer centers reduced their positive margin rates by almost half compared to low volume centers (12.7% versus 21.9% and 11.2% to 19.3%) [55,58]. Furthermore, Oliver et al. found that BOT positive margins were reduced by around 6% when comparing rates from 2011 to 2016 [58]. A systematic review on TORS found that specific procedures were more difficult due to the structural components of the robot, with BOT resections being more likely to undergo conversion to an open approach [59]. As positive margin status is a strong consideration for adjuvant therapy and increased treatment burden, surgeons must be able to counsel patients appropriately on the risks after TORS. Our findings suggest that it may be more challenging to completely clear all margins in BOT OPSCC resections, but the literature suggests that high volume centers can significantly reduce their positive margin rate through more experience.

Even though BOT OPSCC had higher rates of positive margins, our study did not show increased odds of locoregional or metastatic recurrence for BOT OPSCC when compared to tonsillar OPSCC. In surgical oncological care, positive postoperative margins have been widely accepted as a prognostic factor for increased locoregional recurrence [60]. This has been endorsed in numerous studies of OPSCC, with patients who had final positive margins experiencing reduced disease-specific survival, recurrence-free survival, and overall survival [39,61,62,63,64]. However, all of these studies did not report HPV tumor status, which has been shown to be a crucial delineator when considering OPSCC outcomes [48]. Iyer et al. compared disease-specific survival based on margin status stratified by HPV positivity [65]. While patients with positive margins had worse survival in HPV-negative cases, there was no difference in survival based on final margins in HPV-positive patients [65]. A more recent but smaller study endorsed these findings of margin status not being a predictive factor in p16 positive tumors [18]. These findings have been further confirmed by Carey et al., who could not find any pathological features associated with locoregional recurrence in p16 positive OPSCC [38]. However, a recent NCBD did endorse positive margin status as a statistically significant prognosticator for risk of death in the univariate analysis, but these findings did not persist into the multivariate analysis [66]. When considering outcomes by subsite, the current literature is minimal. Only one retrospective review comparing intensity-modulated radiotherapy found BOT OPSCC to have a higher metastatic recurrence rate, while having similar locoregional control compared to tonsillar OPSCC [67]. As our included patients had a pooled proportion of 92.3% p16/HPV-positive malignancies, this study endorses the current literature and deintensification trends that the increased positive margins in BOT tumors may not be an essential variable when considering locoregional and metastatic recurrence.

Regarding rates of postoperative hemorrhage, our study did not find any statistical differences between tonsillar and BOT OPSCC (10.7% and 8.8%). These rates are similar to postoperative hemorrhage in all cases after TORS, which is reported to be between 1.5% and 13.0% [68,69,70,71,72]. As postoperative hemorrhage after TORS can cause mortality, decreasing the risk factors for this complication is crucial to improving surgical care [71]. The literature is limited in directly comparing hemorrhage rates between tonsil and BOT. One study endorsed no differences in both univariate and multivariate analyses [73], but two other studies, including one national database, found that tonsillar OPSCC had higher odds of experiencing postoperative hemorrhage when compared to BOT [74,75]. Neither study had complete explanations for why tonsillar OPSCC had higher hemorrhage rates but argued that it could be due to an increased likelihood that T3 or T4 tumors of the tonsils were being removed with TORS, while T3 and T4 BOT OPSCC would almost exclusively undergo an open approach [74,75]. Since T3 and T4 tumors involve the larger arteries around the tonsillar fossa, theoretically, there could be a higher chance of hemorrhage [74,75]. One of these studies felt that it was more multifactorial secondary to platelet dysfunction, antithrombotic medications, or hepatic insufficiencies [75]. While our study endorsed no differences, the BOT confidence interval was very wide (1.5% to 21.3%), secondary to only three studies with extractable data. Therefore, we feel that this finding should be interpreted with caution and should be used as a hypothesis-generating finding for future research.

The most significant limitation of this study was the lack of subgroup analyses. As most of the studies included in this systematic review looked at other outcomes of TORS after OPSCC, only a few direct comparisons of tonsillar versus BOT OPSCC could be extracted. Therefore, unfortunately, we could not separate patients by their adjuvant therapy status for each subsite. As radiotherapy or chemoradiotherapy has been supported for tumors with positive margins [76,77], these treatments may have influenced the nonsignificant differences in locoregional and distant control seen between tonsillar and BOT OPSCC, even though BOT OPSCC had higher rates of positive margins. Only two studies reported differences in adjuvant therapy between subsites, with Fradet et al. finding no differences in the rate of adjuvant therapy, whereas Persky et al. found that BOT OPSCC underwent more chemoradiation secondary to more positive margins [12,40]. In looking at all the subsites combined, a few studies have endorsed TORS with adjuvant therapy to be superior to TORS alone, particularly when considering locoregional control [16,38]. However, the literature is not consistent, with two studies showing excellent locoregional (3.0% to 3.3%) and distant control (0.0% to 8.4%) in patients with HPV-positive OPSCC undergoing single modality TORS therapy, regardless of margin status [78,79]. Furthermore, more studies have shown that both negative margin status and adjuvant therapy do not improve recurrence and survival and that salvage therapy can be just as effective for those patients with recurrences [39,42,43,80]. Nichols et al. even said that, because of the high survival rate in HPV OPSCC after TORS, margins and adjuvant therapy most likely do not contribute to overall outcomes. As adjuvant radiotherapy and chemoradiotherapy have been shown to significantly worsen quality of life compared to TORS [81,82], more research is needed to fully understand the influence of adjuvant therapy on recurrence in OPSCC with positive margins.

Furthermore, we could not separate patients with more significant disease, such as T3 and T4 staging, an elevated Charlson comorbidity index, or extranodal extension. In particular, for the postoperative hemorrhage analysis, we could not separate patients on anticoagulation or antiplatelet medications, which have shown to be a significant risk factor for post-TORS hemorrhage [71,74]. Furthermore, this study was limited by the heterogeneity between studies. Some studies reported on their first implementations of TORS at their institution, while others performed recent retrospective reviews after a decade of use. As increased experience with TORS can decrease postoperative positive margins, the length of institutional experience with TORS may have been a confounding factor in these analyses [58]. Moreover, while most studies included upfront TORS for OPSCC, some included salvage procedures that were unable to be factored out of the analysis, therefore, potentially elevating rates of positive margin, recurrence, and hemorrhage for that particular study. In addition, positive margins were not classified uniformly, with some studies using the typical 5 mm mark for clear margins, while others classified 2 mm or close margins as clear. Follow-up varied significantly with the studies, with some studies only including an average of ten months of follow-up, with others including up to forty months. This factor could decrease the studies’ measurement of locoregional or metastatic recurrence with short follow-up compared to those with extensive follow-up. Lastly, the fact that all studies were retrospective case series limits how this analysis translates to the general population. Therefore, a prospective observational trial using standardized definitions of a positive margin and follow-up would better elucidate differences in tonsillar and BOT OPSCC outcomes after TORS.

## 5. Conclusions

In this systematic review comparing tonsillar and BOT OPSCC, a higher rate of positive margins was seen in BOT OPSCC. However, there was no difference in the locoregional and metastatic recurrence rates between these subsites. In addition, postoperative hemorrhage rates did not statistically differ. High heterogeneity between the studies limited the ability to perform clinically relevant subgroup analyses to identify which particular tonsillar or BOT OPSCC patients would be at a higher risk of these poor outcomes. Therefore, future research to identify these risk factors in patients with each tonsillar and BOT OPSCC is warranted to improve the utility of TORS and oncological care.

## Figures and Tables

**Figure 1 cancers-14-03837-f001:**
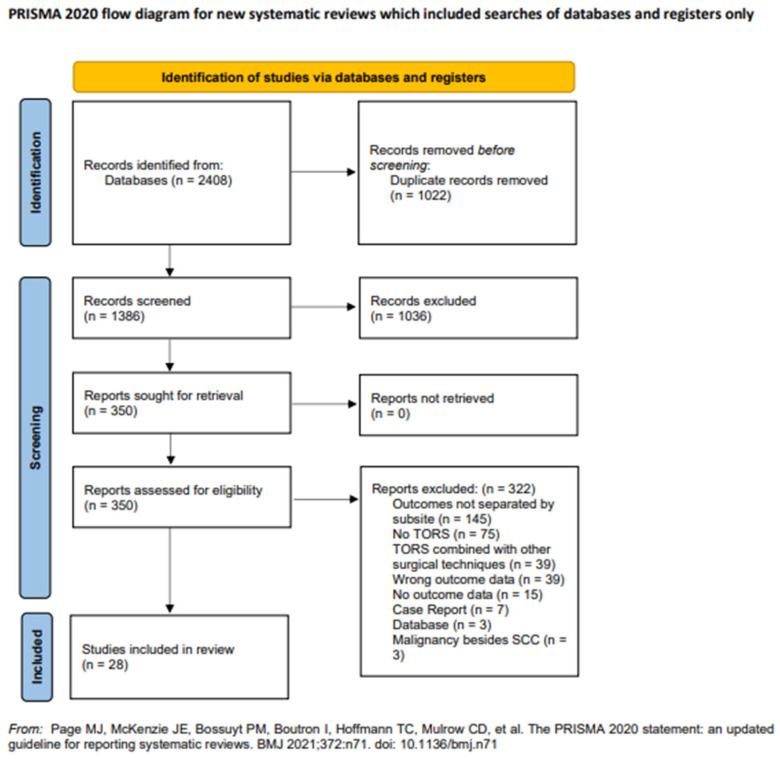
PRISMA flowchart of study selection.

**Figure 2 cancers-14-03837-f002:**
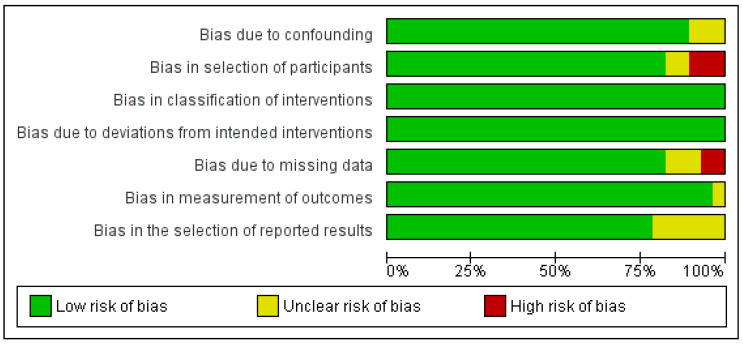
Risk of bias assessment of selected studies.

**Figure 3 cancers-14-03837-f003:**
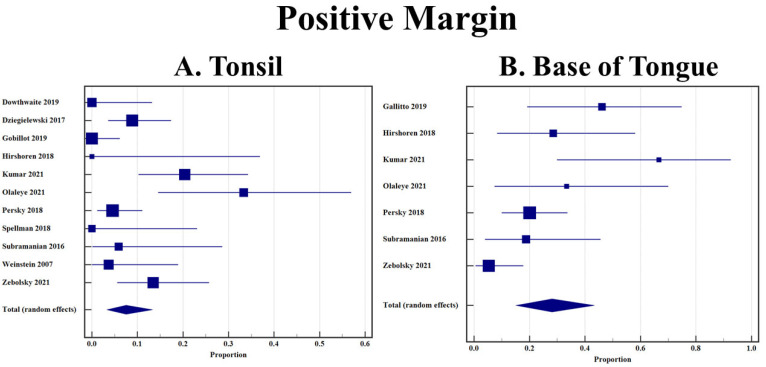
(**A**) Rate of positive margin in tonsillar OPSCC after TORS; (**B**) Rate of positive margin in base of tongue OPSCC after TORS.

**Figure 4 cancers-14-03837-f004:**
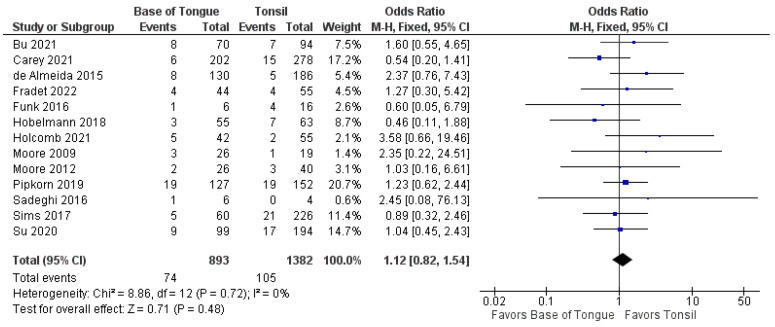
Comparison of total recurrence between tonsillar OPSCC and base of tongue OPSCC.

**Figure 5 cancers-14-03837-f005:**
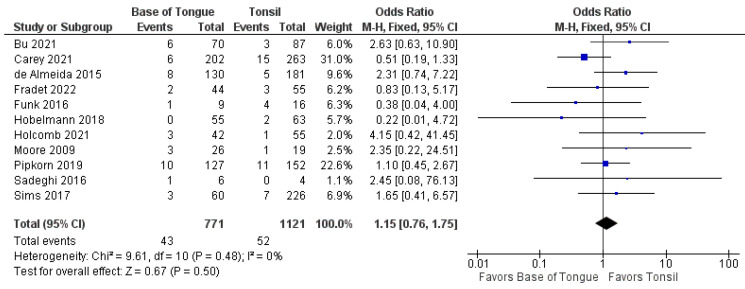
Comparison of locoregional recurrence between tonsillar OPSCC and base of tongue OPSCC.

**Figure 6 cancers-14-03837-f006:**
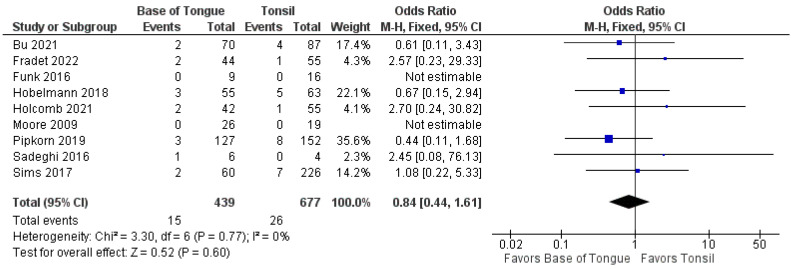
Comparison of metastatic recurrence between tonsillar OPSCC and base of tongue OPSCC.

**Figure 7 cancers-14-03837-f007:**
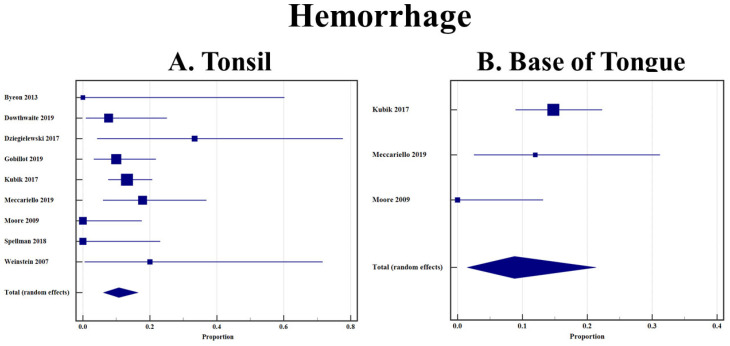
(**A**) Rate of postoperative hemorrhage in tonsillar OPSCC after TORS; (**B**) Rate of postoperative hemorrhage in base of tongue OPSCC after TORS.

**Table 1 cancers-14-03837-t001:** Description of included studies.

Author (Year)	OLE	Study Design	Total Tonsil (N)	Total BOT (N)	P16/HPV (%)	Age Mean or Median, Year (Range)	Male (%)	Never Smoked (%)	Neck Dissection (%)	T3 or T4 (%)	N2 or N3 (%)	Perineural Invasion (%)	Extra-nodal Extension (%)	Lympho-vascular Invasion (%)	Extra-capsular Spread (%)	Adjuvant Radiotherapy (%)	Adjuvant Chemoradiotherapy (%)	Follow-Up Mean or Median, Year (Range)
Bu 2021	4	Case Series	94	70	100.0	60.0 (15.0–85.0)	88.2	NR	NR	3.0	58.6	NR	NR	NR	24.9	42.0	35.5	NR (NR)
Byeon 2013	4	Case Series	4	0	NR	54.0 (38.0–64.0)	75.0	NR	100.0	25.0	100.0	NR	NR	NR	25.0	NR	NR	NR (NR)
Carey 2021	4	Case Series	278	202	100.0	59.1 (NR)	86.7	32.5	NR	7.6	16.1	15.3	30.1	29.2	NR	45.8	37.7	NR (NR)
de Almeida 2015	4	Case Series	186	130	69.4	59.6 (NR)	82.4	33.7	80.8	9.3	48.0	22.3	31.3	26.4	36.7	25.9	17.6	20.0 (1.0–74.0)
Dowthwaite 2019	4	Case Series	26	0	92.3	63.0 (41.0–77.0)	80.8	76.9	26.9	0.0	0.0	NR	NR	NR	NR	11.5	7.7	36.0 (6.0–54.0)
Dziegielewski 2017	4	Case Series	79	0	85.9	55.6 (39.2–78.5)	77.2	15.2	NR	8.9	78.5	31.6	NR	40.5	35.1	30.4	53.2	10.5 (0.0–42.3)
Fradet 2022	4	Case Series	55	44	91.3	59.0 (NR)	85.4	NR	93.2	2.9	40.8	8.7	47.6	50.5	NR	16.5	41.7	31.2 (3.0–9.2)
Funk 2016	4	Case Series	16	9	100.0	58.0 (39.0–91.0)	NR	NR	NR	NR	68.0	12.0	NR	92.0	28.0	NR	0.0	31.5 (4.9–73.1)
Gallitto 2019	4	Case Series	26	19	79.5	54.0 (NR)	95.7	51.1	100.0	4.3	17.4	26.1	97.8	32.5	67.4	100.0	0.0	48.4 (NR)
Gobillot 2019	4	Case Series	58	0	92.7	NR (NR)	89.7	NR	94.8	0.0	8.6	NR	19.0	NR	NR	36.2	24.1	19.5 (6.2–86.0)
Hirshoren 2018	4	Case Series	9	18	62.5	63.7 (28.0–87.0)	62.9	NR	NR	4.2	21.7	NR	NR	NR	NR	NR	NR	NR (NR)
Hobelmann 2018	4	Case Series	63	52	100.0	58.0 (38.0–87.0)	87.1	44.8	NR	NR	NR	NR	43.1	NR	NR	32.8	58.6	30.0 (8.0–82.0)
Holcomb 2021	4	Case Series	55	42	100.0	60.9 (NR)	82.8	45.9	NR	2.0	46.4	3.5	89.9	18.7	NR	0.0	0.0	28.5 (6.0–121.0)
Kubik 2017	4	Case Series	114	122	82.6	59.0 (NR)	81.1	NR	NR	10.2	NR	NR	NR	NR	NR	NR	NR	NR (NR)
Kumar 2021	4	Case Series	49	9	100.0	59.8 (40.0–78.0)	76.3	NR	NR	1.7	39.7	NR	NR	NR	NR	NR	NR	NR (NR)
Meccariello 2019	4	Case Series	28	25	55.0	NR (NR)	NR	NR	NR	12.2	24.5	NR	NR	NR	NR	33.3	36.7	30.3 (NR)
Moore 2009	4	Case Series	19	26	NR	57.0 (38.0–88.0)	88.9	66.7	95.6	24.4	68.9	NR	NR	NR	NR	17.8	55.6	12.3 (1.0–16.0)
Moore 2012	4	Case Series	40	26	71.7	55.2 (36.0–80.0)	89.4	50.0	100.0	9.7	74.2	3.0	NR	12.1	56.1	21.2	62.1	36.0 (24.0–45.0)
Olaleye 2021	4	Case Series	21	22	87.8	60.5 (NR)	82.6	NR	61.2	10.6	10.0	NR	NR	NR	NR	0.0	83.7	NR (NR-54.0)
Persky 2018	4	Cohort	89	51	88.5	58.0 (NR)	80.0	34.5	NR	NR	36.4	16.8	NR	16.9	NR	NR	NR	NR (NR)
Pipkorn 2019	4	Cohort	133	108	100.0	54.0 (27.0–83.0)	86.0	47.3	100.0	22.6	NR	11.0	NR	31.3	78.4	41.9	34.1	33.0 (5.0–65.0)
Sadeghi 2016	4	Case Series	4	6	80.0	59.8 (NR)	80.0	NR	NR	20.0	0.0	NR	NR	NR	NR	0.0	100.0	NR (NR)
Sims 2017	4	Case Series	19	5	100.0	59.6 (NR)	82.6	NR	NR	26.1	56.5	NR	NR	NR	NR	17.4	30.4	13.2 (0.0–74.4)
Spellman 2018	4	Case Series	14	0	100.0	57.6 (32.0–72.0)	78.6	78.6	100.0	0.0	0.0	7.1	NR	0.0	NR	7.1	NR	28.0 (1.0–56.0)
Subramanian 2016	4	Case Series	17	16	90.3	56.5 (NR)	73.5	38.2	85.3	12.1	52.9	NR	NR	NR	51.7	20.6	50.0	NR (NR)
Su 2020	4	Case Series	194	99	100.0	60.0 (28.0–86.6)	84.9	NR	NR	3.2	48.6	NR	NR	NR	NR	0.0	NR	20.0 (NR)
Weinstein 2007	4	Case Series	27	0	NR	NR (NR)	92.6	NR	96.3	22.2	37.0	7.4	NR	NR	NR	33.3	55.6	NR (NR)
Zebolsky 2021	4	Case Series	52	38	100.0	63.0 (36.0–87.0)	83.1	47.1	100.0	2.2	70.6	8.8	19.9	25.0	NR	NR	NR	NR (NR)

OLE = Oxford Level of Evidence; CC = case-control; CS = case series; NR = not reported; N = number of patients; BOT = base of tongue.

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
