# Peer review of "Transoral Robotic Surgery for Oropharyngeal Squamous Cell Carcinoma of the Tonsil versus Base of Tongue: A Systematic Review and Meta-Analysis"

_cancers, 2022, doi:10.3390/cancers14153837_

Round 1
Reviewer 1 Report
Dear Authors,
The paper is very accurate. The direction of the manuscript is clear. The title clearly express the aim of the study and it does not contain any unnecessary description. The abstract can stand alone: it is a concise and sharp summary of the aims, key methods, important findings and conclusions. The introduction clearly summarize the current state of the topic and it is consistent with the rest of the manuscript. It addresses the limitations of the current knowledge in this field, underlying why the study was necessary. The research question is clear and appropriate.
The methods section is very well accomplished. The study design and methods are detailed and appropriate for the research method. There are enough details to repeat the study. How the research was conducted is straightforward: the PRISMA protocol is correctly used and the search strategies are clearly presented as well as the MeSH terms. The Risk of Bias is also calculated by ROBINS-I tool. Data analysis and statistical analysis methods are definite. The results are accurate and match the methods. All the data have been included and are consistent with the data in the figures and tables. The findings are properly described and compared to those available in literature. Their implications for future research, potential applications and limitations are discussed. The references are satisfactory. The search terms and inclusion/exclusion criteria are designated and they ensure all the relevant articles are included.
Reviewer 2 Report
The authors aimed to perform a systematic review of the literature to analyze tonsil or base of tongue (BOT) oropharyngeal squamous cell carcinoma (OPSCC) margin status and recurrence rates after transoral robotic surgery (TORS); secondarily, they also aimed to see if complication rates differed between the subsites. The study is well performed. Here some minor suggestions to improve the quality of the manuscript:
ABSTRACT
- The aim of the study should be clearly reported.
INTRODUCTION
- In my opinion this section could be improved reporting more evidence regarding, OPSCC etiology and pathogenesis, in particular by noting whether some types of OPSCC are more aggressive than others (eg HPV related) regardless of the therapeutic approach adopted. This would be useful to highlight how new techniques could be useful for improving medium- and long-term survival.
MATERIALS AND METHODS
- Why did you choose to also include case series (with at least 4 patients)?
